# Unify Graph Learning with Text:
# Unleashing LLM Potentials for Session Search

## ABSTRACT

Session search involves a series of interactive queries and actions by a user to fulfill a complex information need. Current strategies typically prioritize sequential modeling for deep semantic understanding, often overlooking the graph structure in interactions. On the other hand, while some approaches focus on capturing structural behavior data, they use a generalized representation for documents, neglecting the nuanced word-level semantic modeling. In this paper, we propose a model named Symbolic Graph Ranker (SGR), which aims to take advantage of both text-based and graph-based approaches by leveraging the power of recent Large Language Models (LLMs). Concretely, we first introduce a method to convert graph structure data into text using symbolic grammar rules. This allows integrating session search history, interaction process, and task description seamlessly as inputs for the LLM. Moreover, given the natural discrepancy between LLMs pre-trained on textual corpora, and the symbolic text we produce using our graph-to-text grammar, our objective is to enhance LLMs' ability to capture graph structures within a textual format. To achieve this, we introduce a set of self-supervised symbolic learning tasks including link prediction, node content generation, and generative contrastive learning, to enable LLMs to capture the topological information from coarse-grained to fine-grained. Experiment results and comprehensive analysis on two benchmark datasets, AOL and Tiangong-ST, confirm the superiority of our approach. Our paradigm also offers a novel and effective methodology that bridges the gap between traditional search strategies and modern LLMs[1].

## CCS CONCEPTS

• **Information systems** → **Retrieval models and ranking.**.

**ACM Reference Format:**
Anonymous Author(s). 2024. Unify Graph Learning with Text: Unleashing LLM Potentials for Session Search. In *Proceedings of Make sure to enter the correct conference title from your rights confirmation emai (ACM WWW '24)*. ACM, New York, NY, USA, 10 pages. https://doi.org/XXXXXXX.XXXXXXX

## 1 INTRODUCTION

To address intricate information requirements, users often engage in multiple rounds of interaction with search engines to obtain results that better align with their search intent. This series of user

[1]https://anonymous.4open.science/r/SGR-A6E5/

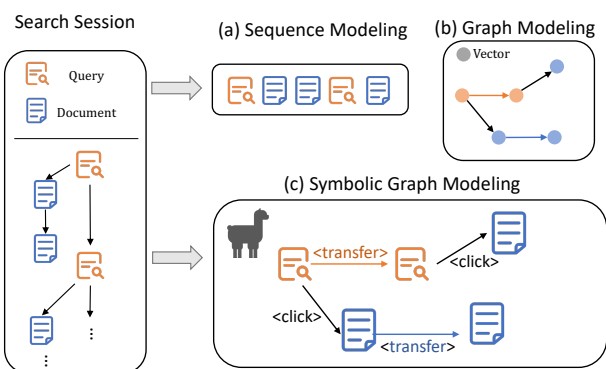

**Figure 1: Comparing paradigms for session search: (a) sequential, (b) graph-based, and (c) our symbolic sequence modeling. Our method takes advantage of the benefits of both sequential modeling (for enhanced semantic encoding) and graph modeling (to capture structural user behavior).**

activities such as issuing queries and clicking on items within a compact time interval is often denoted as a *search session*. The contextual information in search sessions, including sequences of queries and user click behaviors, can be harnessed to enhance the efficacy of search systems [8, 34, 36, 51, 58].

A series of studies view a user's search session as a sequential behavior. In this approach, all the queries and documents in a session are alternatively concatenated using a pre-trained language model, such as BERT [14], to produce the ultimate search results [6, 34, 56], as shown in Figure 1(a). These techniques are adept at capturing the semantic meaning of user queries since the information flows throughout the input at a fine-grained word level. However, these studies often overlook that a search session is a dynamic interaction process with rich user engagement data, not just mere linguistic text. More recently, another line of research seeks to better harness the structural data within the search session through graph modeling. In this approach, the queries and documents within a session form a heterogeneous graph [26, 43] as in Figure 1(b). Nevertheless, the nodes in the graph offer one overall and coarse representation of each document or query, as they overlook the detailed nuances at the word level.

To address the aforementioned challenge and merge the strengths of both techniques, an intuitive way is to transform the heterogeneous structural information into text that can be understood by language models. In this way, the word-level information can fully flow throughout the language model while keeping the graph structure information. However, this process requires the model to have profound language comprehension and reasoning capabilities, for capturing the graph's structural nuances from text and for assessing the relevance between the query and the document - a challenge

that might hinder previous research. Fortunately, with the advent of LLMs, their impressive proficiencies have been showcased across numerous NLP and IR tasks [32, 38, 39]. They provide deep contextual insights, precise semantic understanding, and hold great promise in modeling content across various modalities [53, 54]. Thus, we are optimistic about leveraging LLMs to interpret graph structures linguistically, enabling a comprehensive exploration of both semantic and structural information within a search session.

Concretely, we first create a heterogeneous graph, distinguishing between *query* and *document* as the primary node types. To capture the diversity of user interactions, we also integrate three distinct edge types *click on*, *query transition*, and *document transition*. Then, to represent the rich heterogeneous information from a search session, we transform the explicit structural details of the graph into text conforming to a specific symbolic textual format. We integrate the graph and task instructions into our prompt design, which serves as input to the LLM. The prediction distributions of the answer tokens are used as relevance probabilities for ranking. In this way, we formulate the session search task as the link prediction between the document node and the query node of a session graph, all in text format. Note that while LLMs are pretrained on pure text, we come up with new symbolics to represent graphs. Hence, it is necessary to augment the LLM's comprehension of these symbols. Correspondingly, we propose a set of pre-training tasks including link prediction, generation of node text attributes, and generative contrastive learning with graph augmentation. These tasks reflect the topological information of a session graph from coarse-grained to fine-grained, heuristically guiding the LLMs to understand the heterogeneous session graph structure. By pre-training LLMs on in-domain datasets, we equip them with domain-specific knowledge. This becomes particularly useful when a query or a document recurs across multiple sessions, as this global graph information is stored within the LLM's parameters. Consequently, these LLMs not only capture an inter-session perspective but also enhance their comprehension of the intra-session context.

Experiment results on two public search log datasets, AOL and Tiangong-ST, show that our proposed method outperforms the existing methods with relatively little training data. We also conduct extensive experiments to verify the effectiveness of our symbolic graph representation, and showcase how our symbolic learning tasks enhance LLMs' comprehension of graphs.

Our contributions in this paper can be summarized as follows:

- We aim to integrate structural information in session search with textual data, ensuring that both semantic meaning and topological knowledge from the search session are fully explored and utilized to yield better search results.
- To accomplish this, we harness the capabilities of LLMs, converting graph data into text using a series of symbolic rules. Recognizing the disparity between LLMs and graph-based symbols, we devise a range of self-supervised pretraining tasks to better adapt the LLM for our purpose.
- Our experimental findings, derived from two widely recognized search log datasets (AOL and Tiangong-ST), indicate that our proposed technique surpasses existing methods, especially when training data is limited.

## 2 RELATED WORK

### 2.1 Session Search

Contextual information in sessions is considered conducive to infer users' search intent, providing retrieval results that better align with users' information needs. Early studies extract statistical and rule-based features from users' search history so as to better characterize their search intent [37, 44, 46]. With the development of deep learning approaches, a series of works have emerged that model user behavior sequences to obtain semantically dense representations for session search tasks. For instance, Ahmad et al. [1] utilize a hierarchical neural structure with RNNs to model the session sequence and achieve competitive performance in both document ranking and query suggestion. Taking one step further, the attention mechanism is introduced to existing RNN-based architecture and yields better results [2]. As Pre-trained Language Models have demonstrated their capabilities in various NLP and IR tasks, employing a PLM as backbone has become a new paradigm that treats each search session as a natural language sequence [6, 34, 43, 56].

Recent works suggest that modeling search sessions as sequences may ignore the topological interactions between queries and documents, while the session history can be regarded as a graph for interaction modeling. For example, Ma et al. [26] regard session search as a graph classification task on a heterogeneous graph that represents the search history in each session, and Wang et al. [43] propose a heterogeneous graph-based model with a session graph and a query graph. However, previous Graph Neural Network (GNN)-based studies tend to solely focus on the session structure while neglecting the importance of node semantic modeling, where they only use one vector to represent the whole document in the interaction process. In our work, we explore the potential of LLMs to integrate the benefits of the two approaches, i.e., modeling the nuance of semantic meaning as well as the user behavior structure. Concretley, we flatten the session graph in structural language into prompts and design symbolic pre-training tasks to help the LLM understand and reason over the graph.

### 2.2 Pre-training on Graphs

To enable more effective learning on graphs, researchers have explored how to pre-train GNNs for node-level representations on unlabeled graph data. Inspired by pre-training techniques in natural language processing [21] and computer vision [50], recent studies have been proposed to pre-train GNNs with self-supervised information [47]. This approach aims to tackle the issue of limited labels by employing self-supervised learning on the same graph [20], or to bridge the disparity in optimization objectives and training data between self-supervised pre-training activities and subsequent tasks [17, 25]. The representative tasks include link prediction, node classification, and contrastive learning. etc. For instance, Hu et al. [19] introduce a method based on graph generation factorization to guide the foundational GNN model in reconstructing both the attributes and structure of the input graph, and Qiu et al. [33] put forward a contrastive pre-training model devised to capture universal and transferable structural patterns from multiple input graphs. Different from previous works that achieve these tasks in vector space, we transform the task into text format by a set of symbolic grammars.

## 2.3 Large Language Model For Graph Learning

Existing works have demonstrated the outstanding performance of LLM in natural language processing tasks. Recent studies suggest the use of graph data to create heuristic natural language prompts, aiming to augment the proficiency of large language models. For example, in the field of job recommendations, Wu et al. [45] propose using a meta-path prompt constructor to fine-tune a large language model recommender to understand user behavior graphs. In the domain of molecular property prediction, Zhao et al. [52] introduce a unified language model for both graph and text data, eliminating the need for additional graph encoders to learn graph structures. Through instruction pre-training on molecular tasks, the model effectively transfers to a wide range of tasks. Different from previous works, our approach structures historical conversation graphs into natural language prompts to fully exploit the advantages of LLM.

## 3 METHODOLOGY

### 3.1 Task Formulation

Before introducing our proposed methodology, we first state some notations and briefly formulate the task of session search. We denote the historical queries of a user's search session as $Q = \{q_1, q_2, \ldots, q_M\}$, where each query is the text that the user submitted to the search engine and has been ordered by their issued timestamp. $M$ is the history length Given the query $q_i$, its candidate documents list is denoted as $D_i = \{d_{i,1}, d_{i,2}, \ldots, d_{i,N}\}$, and each document has a binary click label $y_{i,j}$, indicating if user clicks the document. The session search task aims to re-rank the candidate document set $D_i$ considering the in-session contextual information and the issued query $q_i$. In our paper, we denote the session context as the sequence of the historical queries, the clicked documents and the current query. Formally, the session context $S_i$ of $q_i$ is formulated as $\{(q_1, D_j^+), (q_2, D_j^+), \ldots, (q_{i-1}, D_j^+), q_i\}$, where $D_i^+ = \{d_{i,j} \mid y_{i,j} = 1\}$ is the clicked documents of query $q_i$.

### 3.2 Overview

Our model is illustrated in Figure 2. We first establish a heterogenous session graph that stores the user query and interaction information, which is then translated into symbolic text following our specific graph-to-text symbolic grammars. This symbolic depiction, along with the task description, is directly fed into the LLM to generate the ultimate search result. To enhance the LLM's understanding of the symbolic, we come up with three sub-tasks centered around the symbolic text. These sub-tasks require comprehension of the existing graph structure, thus eliminating the need for additional annotations. Finally, we formulate the document ranking task in the search session as a link prediction task, which is also based on the symbolic prediction token.

### 3.3 Session Graph Construction

#### 3.3.1 Graph Schema.
A search session includes user queries, candidate documents, and user click activities. Intuitively, the queries and documents can be regarded as nodes in a graph, with an edge forming between a query-document pair when a user selects the document for a given query. Beyond this, transitions also exist within queries and documents themselves. As such, a search session naturally fits a heterogeneous graph scenario, where multiple sessions sharing common documents and queries weave into a comprehensive global graph.

Formally, the definition of a graph is $G(V, E)$, where $V, E$ denote the sets of the nodes and edges respectively. Our graph is heterogeneous since it contains multiple types of nodes and edges. In the following, we detail the process of constructing our behavior graph for each session. We consider both queries and documents as nodes, i.e., $V = \{q_1, ..., q_i, d_{1,1}, ...d_{i,j}\}$. For these nodes, we define three types of edges:

**Query-query.** Users typically engage in multiple interactions with a search engine to meet their evolving and ambiguous information needs. Consequently, understanding the transition between queries can be beneficial for discerning user search intent. Previous work [43] propose to connect all query pairs in the same session, where a previous query will also be connected to all future queries. Herein, we assume that linking all query pairs could potentially dilute the distinct progression of user intents and introduce unnecessary noise into the session. Therefore, we opt for a more streamlined approach, connecting only adjacent queries denoted as *query transition*, e.g., $e_i = (q_j, q_{j+1})$.

**Query-document.** In large-scale search logs, click-through data is often leveraged as an indicator of the relevance between queries and documents [43]. As a result, we consider the click relation *click on* between queries and their clicked documents, e.g, $e_i = (q_j, d_{j,k})$, where $y_{j,k} = 1$. Liu et al. [23] link queries to their top returned results to enrich relevance signals, addressing the sparsity of click-through in comprehensive search logs. However, we assume this approach might undermine the significance of the click relationship and inadvertently incorporate unrelated documents. To provide more nuanced signals to the model, we resort to the pre-training tasks which will be introduced in §3.4.

**Document-document.** Until now, the constructed edges are centered around the queries. Nevertheless, within the context of session search, documents also hold significant importance. While queries are typically brief and ambiguous outlines, documents provide detailed and specific information related to the query. We believe that transitions between clicked documents can also shed light on the evolution of a user's intent, complementing the insights gained from query transitions. Correspondingly, we construct a fully connected graph, named *document transition*, for the clicked document pairs of the given query, exemplified by edges such as $e_i = (d_{j,k}, d_{j,l})$, where $y_{j,k} = 1$ and $y_{j,l} = 1$. Note that the query-query and query-document edges are asymmetrical, each bearing distinct implications. Conversely, the document-document edges between documents are symmetrical, owing to their shared attribute.

#### 3.3.2 Symbolic Graph Construction.
Traditional works use neural networks for graph modeling [27, 40]. However, the dense vector format is not easily understood by language models. Therefore, our objective is to transform the graph structure into a task-specific symbolic language that's understandable by LLMs. The benefits are twofold. First, LLMs are renowned for their profound contextual insights and accurate semantic understanding, attributes invaluable for text-based session search. Second, symbolic inference ensures transparency and trustworthiness; the reasoning is grounded in

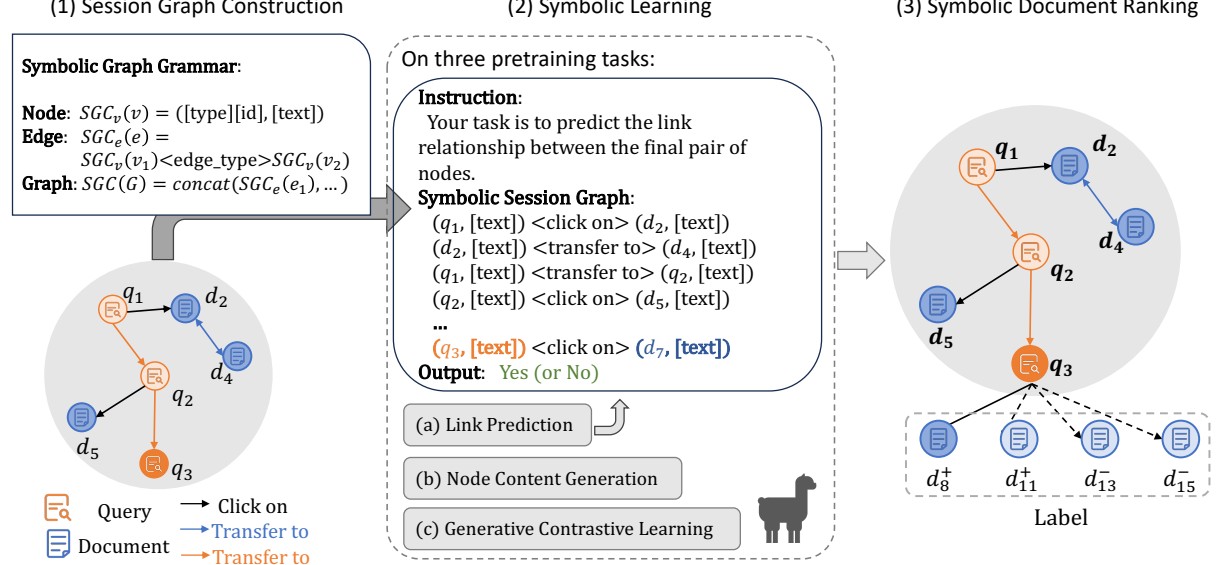

**Figure 2: Overall architecture of our model, which consists of three parts: (1) Session Graph Construction: this part organizes the interaction process as a heterogeneous global graph. (2) Symbolic Learning: we design three self-supervised subtasks to enhance the understanding of LLM on the symbolic representation of the session graph. (3) Symbolic Link Prediction: the LLM is fine-tuned for the document ranking task reframed as symbolic link prediction.**

symbolically represented knowledge and adheres to well-defined inference rules consistent with logical principles [29].

Formally, we transform the session graph $G$ into symbolic language following setting a few symbolic grammars:

• **Nodes:** the node $v \in V$ is either a query $q_i$ or a document $d_{i,j}$. Each node has its node type, index, and text content. Thus, the symbolic representation of a node can be formulated as:

$$SGC_v(v) = ([\texttt{type}][\texttt{id}], [\texttt{text}]). \qquad (1)$$

For example, a third query asking about the MacBook price is denoted as $(q_3, \texttt{MacBook Price?})$, and the corresponding fifth document candidate is represented as $(d_5, \$1,999)$.

• **Edges:** in our symbolic grammar, we define two kinds of edges. Firstly, we retain the original *click on* relationship to represent the direct interaction between a query and its selected documents. Secondly, we include both the query transition and document transition under the umbrella term *transfer to*. This is because our preliminary experiments indicate that differentiating between query-to-query and document-to-document transitions didn't significantly benefit our model's performance. By unifying these transitions under the term *transfer to*, we not only simplify the graph representation but also ensure a more streamlined interpretation for the LLM.

Formally, for the edge $e$ that links the node $v_1$ with $v_2$, the symbolic representation of an edge can be formulated as:

$$SGC_e(e) = SGC_v(v_1) \texttt{ <edge\_type> } SGC_v(v_2). \qquad (2)$$

For example, an edge between the above query node and the document node is denoted as:

$$SGC_v(q_3, d_5) = (q_3, \texttt{MacBook Price?}) \texttt{ <click on> } (d_5, \$1,999).$$

• **Session Graph:** As we have contained the node information in the symbolic representation of the edge, the translation of the session graph $G(V, E)$ into a symbolic language prompt can be represented as the concatenation of the edges in chronological order.

$$SGC(G) = concat(SGC_e(e_1), SGC_e(e_2), \ldots, SGC_e(e_{|E|})). \qquad$$

An example of the symbolic session graph is in Figure 2.

## 3.4 Symbolic Learning

We've developed a unique set of grammars to convert graph structures into symbolic text. This allows LLMs to interpret and analyze them. To further ensure that LLMs are adept at understanding this symbolic representation, we introduce a series of pre-training tasks tailored to familiarize the LLM with the nuances and intricacies of the transformed text.

*3.4.1 Link Prediction.* Link prediction has traditionally been a cornerstone task in self-supervised graph pretraining [4, 49]. It is based on predicting connections between nodes, leveraging the inherent structure and attributes within a graph. This method not only harnesses the topological patterns of the graph but also serves as an effective approach to capture and represent the latent relationships between nodes in a graph-based model. While previous methods computed the similarity based on the embeddings of each node, we have re-envisioned the task into a textual question format, relying on the capabilities of the LLM to discern the nuances and relationships.

Specifically, edges connecting nodes serve as positive samples, while non-connected edges are treated as negative samples. Given two nodes $v_1$ and $v_2$ that we aim to predict a link for, and the graph

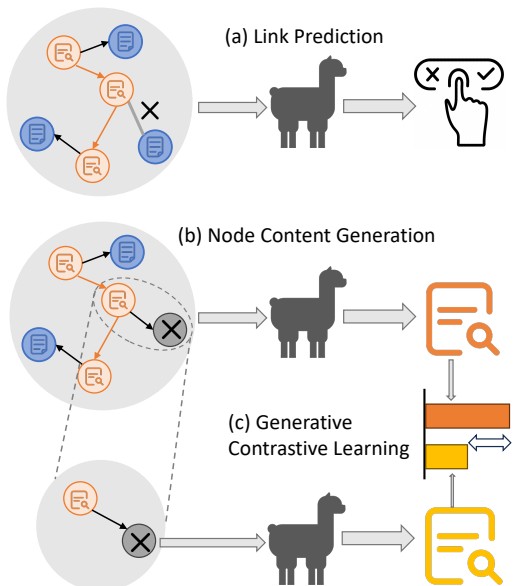

**Figure 3: Three self-supervised symbolic learning tasks to bridge the gap between LLMs and symbolic representations.**

without the target link information $G_{link}$, the input presented to the LLM is structured as follows:

$$X = [\texttt{task\_instruction}]SGC(G_{link}) \tag{2}$$

$$SGC_v(v_1)\texttt{<edge\_type>}SGC_v(v_2). \tag{3}$$

The task instruction example is in Figure 2.

We use $p(X)$ to denote the logits of the answer token predicted by the model, which is considered as the link probability. Thus, the optimization goal is:

$$\mathcal{L}_{link} = -z \cdot \log p(X) + (1 - z) \cdot \log p(X),$$

where $z$ is the link label. If $z$ is 1 (corresponding to the 'yes' class), the loss function aims to minimize $(-\log p(X))$, pushing the predicted probability for the positive class closer to 1. If $z$ is 0 (corresponding to the 'no' class), the loss function aims to minimize the negative log likelihood of the prediction being the negative class, pushing the predicted probability for the positive class closer to 0.

*3.4.2 Node Content Generation.* Traditional graph modeling typically offers just one overall representation for each node [23, 43, 55]. As a result of this simplification, earlier pre-training tasks on graphs have been largely confined to predicting node attributes, often restricted to a handful of class labels. In contrast, our approach emphasizes preserving the concrete semantic meaning of each node. We achieve this by retaining word-level information for both query and document nodes. Building on this foundation, in this study, we elevate the challenge by requiring the LLM to predict the context within each node, be it the query or the document content.

Specifically, as shown in Figure 3(b), we randomly mask a node $v_2$ in the session graph, denoting the resulting graph as $G_{node}$. The

input to the LLM is then given by:

$$X = [\texttt{task\_instruction}]SGC(G_{node}) \tag{4}$$

$$SGC_v(v_1)\texttt{<edge\_type>}. \tag{5}$$

The training objective of the content generation task is to reconstruct the target node content:

$$\mathcal{L}_{node} = -\sum_{i=1}^{|SGC(v_2)|} \log p(SGC(v_2)_i|X, SGC(v_2)_{<i}),$$

where $SGC(v_2)_{<i}$ represents the words in $SGC(v_2)$ preceding the $i$-th word. Note that we let the LLM predict both the content and the index of the target node, drawing inspiration from existing recommendation works that are based on IDs [16].

*3.4.3 Generative Contrastive Learning.* Contrastive learning is a traditional pre-training task for graphs, as highlighted in studies such as [11–13]. The core objective of this approach is to ensure that the representations of adjacent nodes are similar while distancing those of non-adjacent nodes. Drawing inspiration from this, we present a new paradigm, a generative contrastive learning task tailored for symbolic graph representation. The basic concept is to emphasize the LLM's awareness of the session history. As a result, this method could furnish models with a deeper understanding of context, allowing them to adapt more efficiently to evolving graph structures over time.

Specifically, as in Figure 3(c), we consider two distinct scenarios for inputs: In the first scenario, the LLM predicts the content of the target node with access to the search history, represented as $X = SGC(G)SGC_v(v_1)\texttt{<click on>}$. In the second scenario, the LLM lacks this access, denoted by $X_s = SGC_v(v_1)\texttt{<click on>}$. Our objective is for the performance with history to surpass that of the model without it.

$$\mathcal{L}_{contras} = \sum_{i=1}^{|SGC_v(v_2)|} \log p(SGC_v(v_2)_i|X, SGC_v(v_2)_{<i}) \tag{6}$$

$$- \sum_{i=1}^{||SGC_v(v_2)|} \log p(SGC_v(v_2)_i|X_s, SGC_v(v_2)_{<i}). \tag{7}$$

## 3.5 Symbolic Document Ranking

Our ultimate objective for session search is to provide a sequence of related documents. This approach shares similarities with the link prediction task discussed in Section § 3.4.1, but it differs in that it involves returning either a single result or multiple results. Consequently, the optimization function also varies. For a given query node $q$ and a candidate document $d_j$ from the candidate sets within a user session graph $G$, the input presented to the LLM is formulated as:

$$X_j = [\texttt{task\_instruction}]SGC(G) \tag{8}$$

$$SGC_v(q)\texttt{<click\_on>}SGC_v(d_j). \tag{9}$$

The logits of the 'yes' answer token $p(X_j)$ is regarded as the ranking score of document $d_j$. To optimize the model, we employ the negative log-likelihood loss for learn-to-rank as follows:

$$\mathcal{L}_{rank} = -\log \frac{e^{p(X_+)}}{\sum_{X_j} e^{p(X_j)}},$$

where $X_+$ denotes the related positive documents. This loss tries to push the positive document's score higher than those of other documents.

**Table 1: The statistics of the two datasets used in our paper. The number in parentheses is the average number of relevant documents.**

| AOL | Training | Validation | Test |
|---|---|---|---|
| # Sessions | 219,748 | 34,090 | 29,369 |
| # Queries | 566,967 | 88,021 | 76,159 |
| Avg.# Query per Session | 2.58 | 2.58 | 2.59 |
| # Candidate per Query | 5 | 5 | 50 |
| Avg. Query Len | 2.86 | 2.85 | 2.9 |
| Avg. Document Len | 7.27 | 7.29 | 7.08 |
| Avg. # Clicks per Query | 1.08 | 1.08 | 1.11 |
| **Tiangong-ST** | **Training** | **Validation** | **Test** |
| # Sessions | 143,155 | 2,000 | 2,000 |
| # Queries | 344,806 | 5,026 | 6,420 |
| Avg.# Query per Session | 2.41 | 2.51 | 3.21 |
| # Candidate per Query | 10 | 10 | 10 |
| Avg. Query Len | 2.89 | 1.83 | 3.46 |
| Avg. Document Len | 8.25 | 6.99 | 9.18 |
| Avg. # Clicks per Query | 0.94 | 0.53 | (3.65) |

## 4 EXPERIMENT SETUP

### 4.1 Research Questions

We list four research questions that guide the experiments:

• **RQ1** (See § 5.1): What is the overall performance of SGR compared with different kinds of baselines?

• **RQ2** (See § 5.2): What is the effect of each module in SGR? Is the performance improvement attributed to the symbolic graph representation we propose?

• **RQ3** (See § 5.3): Is our method robust with session lengths?

• **RQ4** (See § 5.4): How does our model scale with data?

• **RQ5** (See § 5.5): How does SGR perform in the pre-training stage?

### 4.2 Dataset and Evaluation Metrics

*4.2.1 Dataset.* Following previous studies [43], we conduct experiments on two large-scale search log datasets, i.e., AOL [30] and Tiangong-ST-click [9].

We use the AOL dataset provided by Ahmad et al. [2]. It contains numerous search logs grouped as sessions. Specifically, five candidate documents are provided for each query in both training and validation sets. In the test set, for each session query, we utilize 50 documents retrieved by BM25 [35] as candidates. Every query in this dataset has a minimum of one corresponding click. When multiple clicked documents exist for a query, we construct the user behavior sequence using the first document from the list.

For the Tiangong-ST dataset, the session data are extracted from an 18-day search log provided by a Chinese search engine, and each query has ten candidate documents. Our setting follows [56]. In training and validation sets, we use the click-through labels as relevant signals. In this test set, only prior queries—excluding the last one—and their associated candidate documents are used. As with the AOL dataset, documents in this test scenario are labeled either 'click' or 'unclick'. The model's objective is to rank clicked documents as highly as possible. Note that queries without any

clicked documents are excluded from testing. The statistics of both datasets are shown in Table 1.

*4.2.2 Evaluation Metrics.* Following the earlier studies, we employ the Mean Average Precision (MAP), Mean Reciprocal Rank (MRR), and Normalized Discounted Cumulative Gain at position $k$ (NDCG@$k$, $k$ = 1, 3, 5, 10) as metrics. All evaluation results are computed by the TREC's official evaluation tool (trec_eval) [42].

### 4.3 Baseline

In our experiment, we compare our methods with two kinds of baselines including (1) *ad-hoc* ranking methods, and (2) *context-aware* ranking methods.

(1) Ad-hoc ranking. These methods focus on the matching between the issued query and candidate documents, neglecting the information from the search context.

• **BM25** [35] is a traditional probabilistic model, which models the relevance of a document to a query as a probabilistic function. We use the pyserini [22] tool in our work to calculate the BM25 scores.

• **MonoT5** [28] is a sequence to-sequence re-ranker that uses T5 to calculate the relevance score. In our paper, we use the trained checkpoint on Ms Marco [3] on HuggingFace.

(2) Context-aware ranking methods These methods employ either sequential modeling to process historical queries or graph-based modeling to represent user behavior.

• **RICR** [5] is an RNN-based method that uses the history representation to enhance the representation of queries and documents on word-level.

• **COCA** [57] pre-trains a BERT encoder with data augmentation and contrastive learning for better session representation.

• **ASE** [7] designs three generative tasks to help the encoding of the session sequence. In contrast to other multi-task approaches that consider only the subsequent query generation as the auxiliary task, it further takes the succeeding clicked document and a similar query as generation targets.

• **HEXA** [43] proposes graph modeling user behavior in session search. It constructs two heterogeneous graphs, a session graph, and a query graph, to capture user intent from global and local aspects respectively.

### 4.4 Implementation Details

We use PyTorch [31] to implement our model. Specifically, LLaMa-7B [41] and BaiChuan-7B [48] are used as the backbone LLMs for AOL and Tiangong-ST, respectively. To facilitate lightweight fine-tuning, we employ LoRA [18] to train our model, which freezes the pre-trained model parameters and introduces trainable rank decomposition matrices into each layer of the Transformer architecture. We adopt AdamW optimizer [24] to train our model. The learning rate is set as 2e-5 with a cosine decay. We train our model by 2 epochs and the batch size is set as 8. Due to computational constraints, we randomly selected 1,000 sessions from the AOL test set for evaluation. All hyperparameters are tuned based on the performance of the validation set. For further implementation details, please refer to our code[2].

---

[2]https://anonymous.4open.science/r/SGR-A6E5/

**Table 2: The overall results of our model and compared baselines on two datasets. The best performance are in bold. "†" and "‡" indicate our model achieves significant improvements over all existing methods in paired t-test with p-value < 0.01 and p-value < 0.05 respectively (with Bonferroni correction).**

| Model | AOL | | | | | | Tiangong-ST-Click | | | | | |
|---|---|---|---|---|---|---|---|---|---|---|---|---|
| | MAP | MRR | NDCG@1 | NDCG@3 | NDCG@5 | NDCG@10 | MAP | MRR | NDCG@1 | NDCG@3 | NDCG@5 | NDCG@10 |
| *ad-hoc ranking:* | | | | | | | | | | | | |
| BM25 | 0.2703 | 0.2799 | 0.1608 | 0.2410 | 0.2693 | 0.3063 | 0.2845 | 0.2997 | 0.1475 | 0.1983 | 0.2447 | 0.4527 |
| MonoT5 | 0.3741 | 0.3856 | 0.2415 | 0.3496 | 0.3816 | 0.4308 | 0.3306 | 0.3447 | 0.1494 | 0.2465 | 0.3315 | 0.4939 |
| *context-aware ranking:* | | | | | | | | | | | | |
| RICR | 0.5506 | 0.5628 | 0.4084 | 0.5442 | 0.5823 | 0.6154 | 0.7472 | 0.7697 | 0.6401 | 0.7450 | 0.7822 | 0.8174 |
| ASE | 0.5667 | 0.5788 | 0.4081 | 0.5732 | 0.6073 | 0.6319 | 0.7410 | 0.7637 | 0.6277 | 0.7381 | 0.7790 | 0.8130 |
| COCA | 0.5649 | 0.5743 | 0.4149 | 0.5655 | 0.6010 | 0.6301 | 0.7481 | 0.7696 | 0.6386 | 0.7445 | 0.7858 | 0.8180 |
| HEXA | 0.5700 | 0.5819 | 0.4165 | 0.5744 | 0.6097 | 0.6372 | 0.7427 | 0.7660 | 0.6352 | 0.7378 | 0.7790 | 0.8141 |
| SGR | **0.5859**† | **0.5972**† | **0.4349**† | **0.5907**† | **0.6225**† | **0.6509**† | **0.7553**‡ | **0.7782**‡ | **0.6503** | **0.7514**‡ | **0.7913**‡ | **0.8239**† |
| Improv. over HEXA | +2.79% | +2.63% | +4.42% | +2.84% | +2.10% | +2.15% | +1.70% | +1.60% | +2.38% | +1.84% | +1.58% | +1.20% |

## 5 RESULTS AND ANALYSIS

### 5.1 Overall Results

Addressing **RQ1**, in Table 2, SGR consistently surpasses other techniques, underscoring our approach's effectiveness. Based on the results, we can make the following observations.

(1) *Context-aware ranking methods consistently outperform ad-hoc models.* While ad-hoc models like BM25 and MonoT5 primarily focus on immediate query-document matching, they overlook the wealth of information embedded in the user's session history. On the other hand, context-aware methods, such as RICR, COCA, ASE, and HEXA, effectively harness sequential or graph-based representations to model user behavior over time. This not only provides a deeper understanding of user intent but also captures evolving search nuances. The superiority of context-aware methods in the results suggests that in a dynamic and interactive search session, understanding the broader context is crucial for achieving higher relevance and accuracy in rankings.

(2) *Our LLM-based SGR model significantly outperforms the state-of-the-art method HEXA (in paired t-test at p-value < 0.01).* While both models integrate session data into their respective graphs, ours does so with better performance. On the other hand, HEXA's heterogeneous graph constructs more edges compared to our model, as discussed in §3.3. Additionally, HEXA introduces both a query and session graph, whereas our model is founded on a singular graph structure. The positive results from our model suggest that LLMs can be effective in graph information modeling when paired with a well-designed pretraining task. Additionally, the potential of our approach is substantial and can grow alongside the evolution of LLM technologies.

### 5.2 Ablation Study

(1) **The effects of various symbolic learning pre-training tasks.** For **RQ2**, we initiated ablation studies to delve into the impacts of different symbolic learning pre-training tasks. The findings of these studies are presented in Table 3. The term 'None' indicates the use of the baseline LLM for document ranking without incorporating our suggested symbolic learning phase.

Notably, the full deployment of SGR strategies yields the best results across all metrics, underscoring the comprehensive strength

**Table 3: Ablation performance of SGR on the AOL dataset with different symbolic learning tasks.**

| | MAP | MRR | NDCG@1 | NDCG@5 | NDCG@10 |
|---|---|---|---|---|---|
| SGR (Full) | **0.5859** | **0.5972** | **0.4349** | **0.6225** | **0.6509** |
| None | 0.5580 | 0.5701 | 0.4050 | 0.5925 | 0.6242 |
| Link | 0.5783 | 0.5907 | 0.4282 | 0.6146 | 0.6443 |
| Node | 0.5747 | 0.5876 | 0.4247 | 0.6097 | 0.6410 |
| Contras | 0.5740 | 0.5863 | 0.4177 | 0.6150 | 0.6398 |
| Link + Node | 0.5846 | 0.5966 | 0.4349 | 0.6206 | 0.6490 |
| Link + Contras | 0.5809 | 0.5925 | 0.4259 | 0.6197 | 0.6469 |
| Node + Contras | 0.5782 | 0.5916 | 0.4294 | 0.6151 | 0.6433 |

**Table 4: Performance of SGR on the AOL dataset with different symbolic strategies. SG denotes 'symbolic graph' and SL denotes 'symbolic learning task'.**

| | MAP | MRR | NDCG@1 | NDCG@3 | NDCG@10 |
|---|---|---|---|---|---|
| SGR (Full) | **0.5859** | **0.5972** | **0.4349** | **0.5907** | **0.6509** |
| SGR w/o SG | 0.5191 | 0.5295 | 0.3594 | 0.5184 | 0.5891 |
| SGR w/o SLT | 0.5580 | 0.5701 | 0.4050 | 0.5612 | 0.6242 |

of a holistic approach. When breaking down combinations, the 'Link + Node' performs the best. However, as strategies are decoupled or used singly, there's a noticeable dip in performance, with the 'None' configuration highlighting the least effectiveness. This gradient in results underlines the criticality of integrative symbolic learning in refining sequence representation and optimizing for superior outcomes.

(2) **The effects of our symbolic graph representation method.** While our SGR demonstrates impressive results, it's crucial to determine whether the improvements come solely from the text in the search history or from the symbolic graph structure. Hence, we design an experiment presented in Table 4.

Our experiment comprises three distinct scenarios: (1) SGR w/o SG (symbolic graph): In this configuration, we omit graph information, which encompasses nodes and edges. Instead, the text is represented by sequences strung together with delimiters. (2) SGR

w/o SL (symbolic learning): While we incorporate the symbolic graph text in this setup, it is directly fine-tuned, bypassing the symbolic learning pre-training phase. (3) Full SGR model: Here, we seamlessly integrate both the symbolic graph input and symbolic learning tasks.

The outcomes indicate that excluding either SG or SL severely impairs the model's performance. Specifically, when we only keep the text while dismissing the graph information (SGR w/o SG), we observe a noticeable decline in MAP and MARR scores. This underscores that the mere inclusion of session history text isn't enough to improve performance. Moreover, pretraining tasks are also important for LLM to understand the graph structure (SGR w/o SLT). This highlights the efficacy of our symbolic graph representations in aiding large language models to grasp and leverage this concept.

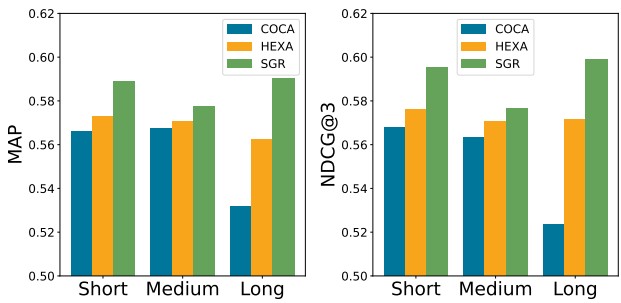

Figure 4: Comparison of MAP and NDCG@3 performance across varying session lengths for COCA, HEXA, and SGR.

### 5.3 Impact of Session Lengths

The session length plays a pivotal role in determining the richness of contextual information, thereby influencing the efficacy of context-aware ranking models. For **RQ3**, to analyze this effect, we categorize the test sessions into three groups: short sessions with a length of 2 or fewer, medium sessions with a length of 3 or 4, and long sessions where the length exceeds 4.

Figure 4 illustrates the superior performance of SGR when compared to multiple baselines, with the $y$-axis representing the MAP score. SGR consistently surpasses all baseline models, highlighting its robustness regardless of session length variations. There is a distinct trend, i.e., the longer the session length, the more pronounced SGR's advantage becomes. This demonstrates the proficiency of the LLM in handling long contexts and intricate behavioral relations. These findings not only validate the efficacy of symbolic graphs in capturing session behaviors but also emphasize the critical role of session search logs in the ranking mechanism.

### 5.4 Impact of Amount of Training Data

Recent research [10, 15] has highlighted the significant influence of the data volume on downstream tasks, such as document ranking in our scenario. To delve deeper into this aspect and address **RQ4**, we trained our model using varying data proportions. Note that due to computational limitations, we only sampled a portion of the corresponding dataset for training and testing.

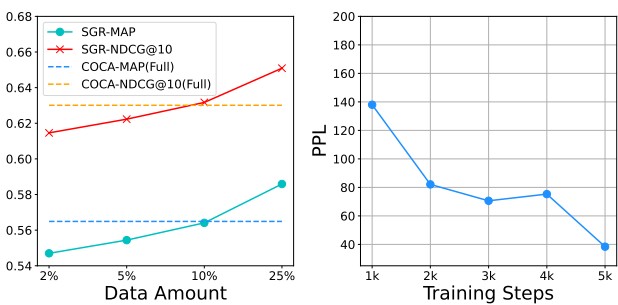

Figure 5: (Left) The performance of SGR with different training data amounts. (Right) Perplexity of SGR along the training process.

In the graph presented on the left in Figure 5, we examine the performance metrics of two models: SGR and COCA, across varying data scales ranging from 2% to 25%. Two key performance indicators, MAP and NDCG@10, were considered. It's evident that our SGR consistently outperforms COCA across all data amounts in both MAP and NDCG@10 metrics. Specifically, SGR outperforms fully-trained COCA with only 10% of training data, which demonstrates the impressive efficiency of the SGR model. The consistent performance elevation of SGR reiterates its robustness and scalability in handling larger datasets, making it a more reliable choice for tasks requiring adaptive search results to data scale variations.

### 5.5 Performance of SGR in Pre-training Stage

For **RQ5**, while our earlier experiments were primarily centered around our core session search task, it's important to note that our model is initially pre-trained on symbolic learning tasks. Hence, apart from the previous experiments that implicitly demonstrate the effectiveness of the pre-training stages, we here examine directly the performance of SGR in the pre-training phase. On the right of Figure 5, we present the perplexity (PPL) scores during the symbolic learning task. Notably, the PPL exhibits a downward trend, indicating substantial improvements in model training. This observation serves as evidence that SGR excels in comprehending symbolic graph grammar during the pre-training phase, successfully capturing the underlying graph structures.

## 6 CONCLUSION

In this paper, we introduced the Symbolic Graph Ranker (SGR), a novel approach that combines the strengths of sequential and structural modeling for session searches, using the prowess of Large Language Models (LLMs). By transforming graph data into text through symbolic grammar rules, and implementing self-supervised pre-training tasks, we effectively bridged the gap between traditional session search methods and LLMs. Our results on AOL and Tiangong-ST datasets validated the superiority of SGR, marking a promising step forward in the realm of session search. Moving forward, we aim to further explore the integration of multi-modal data and refine the self-supervised tasks to enhance SGR's adaptability across diverse search environments.

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
