# OpenReview forum: "Unify Graph Learning with Text: Unleashing LLM Potentials for Session Search"
_ACM.org/TheWebConf/2024/Conference — TheWebConf24_

### Official Review · Reviewer_K28d · 2023-11-20

**Novelty:** 4
**Technical Quality:** 5

**Review:**

This paper focuses on utilizing LLMs for session search by considering both text-based and graph-based approaches.

Pros:
* The idea of converting graph structure data into text to train LLMs is straightforward. Yet it provides a solution to how LLMs can utilize graph data.

* The paper includes various analysis to help understand the proposed method, including ablation studies and effect of training data size.

* The proposed method is evaluated on two public datasets, which facilitates reproducibility.

Cons:
* The baselines are not implemented by LLMs. It's hard to see whether the improvement comes from the LLM itself or by the proposed training strategies.

* It would be useful to include some case studies to gain further understanding of the proposed method.

**Questions:**

NA

**Reviewer Confidence:**

3: The reviewer is confident but not certain that the evaluation is correct

**Scope:**

4: The work is relevant to the Web and to the track, and is of broad interest to the community

---

### Official Review · Reviewer_racG · 2023-11-22

**Novelty:** 5
**Technical Quality:** 5

**Review:**

This work proposes a symbolic graph ranker (SGR) that combines the strengths of sequential and structural modeling for session searches using large language models. It also proposes several symbolic learning tasks for enhancing LLMs' ability of understanding symbolic graphs.

Pros:
1. The presentation of the paper is clear and easy to follow.
2. The idea of utilizing symbolic texts to enable LLMs to understand the graph structure while keeping well semantic understanding is sound.
3. The experiments demonstrate the advantages of the proposed SGR over state-of-the-art baselines.

Cons:
1. The scale and size of the graphs that SGR can handle may be limited by the input length constraints of LLMs. This limitation could restrict the applicability of the proposed graph learning method in tasks involving more complex graph structures, ~~within or~~ beyond session searches.
2. ~~Some additional experiments or case studies on evaluating to what extend does the LLM understand the symbolic graph structures may be interesting.~~ [The authors have addressed my second question.]

With my limited knowledge on symbolic learning, I don't find major weaknesses in this paper.

**Questions:**

NA

**Reviewer Confidence:**

3: The reviewer is confident but not certain that the evaluation is correct

**Scope:**

4: The work is relevant to the Web and to the track, and is of broad interest to the community

---

### Official Review · Reviewer_zJFS · 2023-11-26

**Novelty:** 4
**Technical Quality:** 4

**Review:**

### Summary

This paper introduces the Symbolic Graph Ranker (SGR), a novel approach that integrates symbolic graph representation with large language models (LLMs) to improve session search. The authors conduct extensive experiments on two large-scale search log datasets, AOL and Tiangong-ST, comparing SGR against various ad-hoc and context-aware ranking methods. They also explore the effects of various components of SGR through ablation studies, the impact of session lengths on model performance, and the robustness of SGR in different training data scales.

### Strengths

1. SGR's integration of symbolic graph representation with LLMs is a novel contribution, addressing the challenge of context-aware document ranking in session searches.
2. The paper conducts extensive experiments across multiple datasets and comparison with various baseline models, demonstrating SGR's effectiveness.
3. The ablation studies provide valuable insights into the contributions of different components of the SGR model.

### Weaknesses

1. The paper claims to contribute by integrating semantic meaning with structural information from the search session. However, I believe that references [26] and [43] also merge these two types of information, which potentially diminishes the uniqueness of this paper's contribution.
2. The dual nature of combining symbolic graph representation with LLMs may introduce complexity not only in the initial implementation but also in ongoing maintenance and updates. This complexity could pose challenges for adoption in practical settings, where simplicity and ease of use are often prioritized.
3. In the context of search that is sensitive to user interactions, individuals often explore and engage with numerous documents. This extensive activity can result in overly long sequences, potentially surpassing the processing limits of large language models. The article in question does not adequately address this issue.

**Questions:**

1. How does SGR's performance vary with different LLM backbones?
2. Are there specific types of search queries or sessions where SGR underperforms?
3. How does the computational complexity of SGR compare with baseline models, particularly in real-time search environments?

**Reviewer Confidence:**

3: The reviewer is confident but not certain that the evaluation is correct

**Scope:**

3: The work is somewhat relevant to the Web and to the track, and is of narrow interest to a sub-community

---

### Official Review · Reviewer_deCc · 2023-11-28

**Novelty:** 4
**Technical Quality:** 6

**Review:**

his paper propose to use LLMs to model sessions search, including query reformulation, clicked items.
The session is encoded in a graph and formatted to text for the LLMs to process the intructions.
An LLM of few billions parameters is then finetuned thanks to. LORA on pretraining tasks such as link prediction, query or document content prediction with a constrastive training based on the history.
The application of LLMs to this task is timely and interesting, even if concurrent work that I am not aware of may exists.

Pros:
	- Interesting applications of LLMs to session search.

Cons:
	- A few baseline missing and analysis

On the experiments setting,  I am curious if a fusion of traditional rerankers and session aware model would give, ie a fusion of MonoT5 and HEXA. Furthermore, it Is known that an LLMs will outperform a cross encoder like MonoT5 so a better baseline could be done,
Secondly,   numbers reported from the COCA paper do not match the original paper. Could you clarify this? (COCO NDCG@10 on AOL is 0.60 in this paper while originally 0.616)

What remains unclear to me after reading this paper is whether  one really need a LLMs to do that task. To do so, it could be interesting to report the  zero shot performance of LLMs for that task. As the paper show, an important finetuning is required to make the approach effective. If so,  what would be the performance of a similar finetuning for a BERT or T5 models ( this could be done through the use of special tokens for this models model). Furthermore, could ICL be used rather than this pretraining tasks of finetuning. This remain unclear in the paper despite the significant improvement reported. Similarly, could a good graph neural network  (reusing pretrained text embedding) with appropriate pretraining be as good as an LLMs?  I think this ought to be answer to fully assess the benefit of LLMs for this task.

Finally, there are few missing works in my opinion.
Joint Personalized Search and Recommendation with Hypergraph Convolutional Networks, ECIR'22
GraphPrompt: Unifying Pre-Training and Downstream Tasks for Graph Neural Networks, WWW'23

I am satisfied with the authors response (will increase the technical score). Overall, I think that this is interesting paper for this conference

**Questions:**

- What is your prompt length in the end? Could that scale to document content rather than keyword? For long session and long document?
 - Could you comment on the use and performance of ICL for such model?
 - What is the inference time /efficiency of such method compared to HEXA? Please give indications  on this.

**Ethics Review Flag:**

Yes

**Reviewer Confidence:**

3: The reviewer is confident but not certain that the evaluation is correct

**Scope:**

4: The work is relevant to the Web and to the track, and is of broad interest to the community

---

### Decision · Program_Chairs · 2024-01-22

**Decision:**

Accept

**Comment:**

This is a metareview. It is based on the initial reviews, the author rebuttals, and my own opinion.

 This paper proposes a methodology to encode the structure of a graph into a sequence of symbols using symbolic grammar rules. This enables the authors to explore several scenarios using an LLM which was not previously straightforward to do. Examples include search session history, user interaction data, and task descriptions. Reviewer 1 asks the important question of why we should be using an LLM to model such scenarios, which is a valid question. I don't think the authors attempt to address this metaphysical question. Nevertheless, the authors are able to provide compelling empirical evidence that their approach is effective, and can outperform competitive baselines. The work is timely and relevant, and when considering all of the pros and cons, I believe the work should be accepted. We hope the detailed reviews and discussion will help you address the issues raised in the camera ready version of the paper.